# Study on the Preparation and Optical Properties of Graphene Oxide@Fe_3_O_4_ Two-Dimensional Magnetically Oriented Nanocomposites

**DOI:** 10.3390/ma16020476

**Published:** 2023-01-04

**Authors:** Song Yin, Tiantian Zhang, Yinfeng Yu, Xiaotong Bu, Zepeng Zhang, Junming Geng, Xueling Dong, Haibing Jiang

**Affiliations:** 1Engineering Research Center of Ministry of Education for Geological Carbon Storage and Low Carbon Utilization of Resources, Beijing Key Laboratory of Materials Utilization of Nonmetallic Minerals and Solid Wastes, National Laboratory of Mineral Materials, School of Materials Science and Technology, China University of Geosciences (Beijing), Xueyuan Road, Haidian District, Beijing 100083, China; 2School of Science, China University of Geosciences, Xueyuan Road, Haidian District, Beijing 100083, China; 3School of Chemistry and Chemical Engineering, Beijing Institute of Technology, Beijing 102488, China; 4Lang fang Natural Resources Comprehensive Survey Center, China Geological Survey, Langfang 065000, China

**Keywords:** graphene oxide, magnetic orientation, liquid crystal, magneto-optical materials

## Abstract

In this work, graphene oxide@Fe_3_O_4_ (GO@Fe_3_O_4_) two-dimensional magnetically oriented nanocomposites were prepared through the co-precipitation approach using graphene oxide as the carrier and FeCl_3_·6H_2_O and FeSO_4_·7H_2_O as iron sources. The samples were characterized and tested by X-ray diffraction, a transmission electron microscope, Fourier-transform infrared spectroscopy, a vibrating-specimen magnetometer, a polarized optical microscope, an optical microscope, etc. The effects of material ratios and reaction conditions on the coating effects of Fe_3_O_4_ on the GO surface were investigated. The stable GO@Fe_3_O_4_ sol system was studied and constructed, and the optical properties of the GO@Fe_3_O_4_ sol were revealed. The results demonstrated the GO@Fe_3_O_4_ two-dimensional nanocomposites uniformly coated with Fe_3_O_4_ nanoparticles were successfully prepared. The GO@Fe_3_O_4_ two-dimensional nanocomposites exhibited superparamagnetic properties at room temperature, whose coercive force was 0. The stable GO@Fe_3_O_4_ sol system could be obtained by maintaining 1 < pH < 1.5. The GO@Fe_3_O_4_ sol showed magneto-orientation properties, liquid crystalline properties, and photonic crystal properties under the influence of the external magnetic field. The strength and direction of the magnetic field and the solid content of the GO@ Fe_3_O_4_ sol could regulate the aforementioned properties. The results suggest that GO@Fe_3_O_4_ two-dimensional magnetically oriented nanocomposites have potential applications in photonic switches, gas barriers, and display devices.

## 1. Introduction

Liquid crystal is a phase state between crystals and liquids, which has both the fluidity of a liquid and the anisotropy of a crystal [1]. Typical liquid crystals are based on organic molecules with a rod or disk shape, but inorganic nanoparticles with anisotropic morphology and size can also display the liquid-crystal phase in a dispersed state [2].

Since Zocher [3] discovered liquid-crystal-like materials in the V_2_O_5_ nanoparticle sol system in 1925, inorganic liquid crystals have received considerable attention. Subsequently, liquid-crystal behavior in suspensions of montmorillonite [4], boehmite [5], halloysite [6], graphene oxide [7], and other minerals was observed successively. Inorganic liquid-crystal materials had advantages over organic liquid crystals, including excellent optical and electrical properties, strong thermal stability, and low cost [7,8,9,10]. However, the inorganic liquid crystals formed by these mineral materials had liquid-crystal phenomena or properties whose primary driving force for orientation was gravity or the particle interaction force [7,11]. This led to the sluggish and uncontrollable formation process of the liquid crystal, which made it difficult for the characteristics of liquid crystals to be widely applied.

Magnetic assemblies were considered to be an effective method for rapidly assembling nanomaterials into ordered structures [12,13]. Ge et al. [14] used the highly charged superparamagnetic Fe_3_O_4_ colloidal nanoclusters as substrates preparing magnetically responsive colloidal photonic crystals. Diffraction peaks could be tuned across the entire visible spectrum by simply varying the magnetic field intensity. Subsequently, several applications of the magnetically responsive photonic crystal systems were demonstrated, including anti-counterfeiting devices [15], rewritable test strips [16], and full-color high-resolution printing systems [17]. Similarly, Wang et al. [18] assembled superparamagnetic colloidal nanoparticles into a one-dimensional nanochain structure using an external magnetic field, and a magnetically responsive Bragg reflector with UV–visible light and a diffraction wavelength of 404 nm was successfully fabricated. However, the problem with the aforementioned materials was that magnetic nanospheres had large freedom, which made them struggle to prepare multiple functional materials.

Due to the single magnetic domain structure and high coercivity of metal nanorods, Li [19] and Yuan [20] developed the co-precipitation approach and CPB-assisted (CPB: cylindrical polymer brushes) method, respectively, for assembling Fe_3_O_4_ nanoparticles on silver nanowires and Te nanorods. The results suggested the Fe_3_O_4_-coated nanorods would be oriented by varying the magnetic field. Wang [21] also connected Au nanorods (AuNRs) to Fe_3_O_4_ nanorods whose orientation could similarly be adjusted magnetically. The excitation of AuNR plasmonic modes under normal and polarized light incidence could be regulated by changing the magnetic field direction. However, it was expensive and challenging to industrialize these one-dimensional rare-metal nanowires on a large scale.

To address the issue of the one-dimensional precious metals’ high cost, Fu et al. [22] prepared magnetic nanocomposites composed of palygorskite (Pal) and Fe_3_O_4_ using the co-precipitation approach. In this work, the Pal was employed as the template, and polyethylenimine (PEI) was used as an organic surfactant. It was discovered that PEI could bind Fe^3+^/Fe^2+^ on the Pal and served as a linker for the Fe_3_O_4_ nanoparticle nucleation in situ. The well-dispersed Pal@Fe_3_O_4_ sol system was obtained by adding the dispersant and the sol system exhibited magnetron liquid-crystal properties and Bragg reflections properties. This work served as the foundation for the preparation of Sep@Fe_3_O_4_ one-dimensional magnetic nanocomposites [23] and Mt@Fe_3_O_4_ two-dimensional magnetic nanocomposites [24], both of which displayed excellent magnetically controlled optical properties.

These investigations not only provided fresh research ideas to address the high-cost issue but also proved the development of various magneto-oriented material units. Although PEI coated on the clay minerals’ surface effectively adsorbed Fe^3+^/Fe^2+^, due to the weak interaction between the PEI and clay minerals, there were several issues, such as the uneven coating and the easy shedding of magnetic nanoparticles, which affected the stability of the nanocomposites and the magnetic service life of the materials.

Graphene oxide (GO) is an oxidized carbide with a two-dimensional structure, which is abundant in oxygen-containing functional groups such as carbonyl, carboxyl, and hydroxyl groups, etc. [25]. The structure and composition are favorable for the preparation of uniform and stable GO@Fe_3_O_4_ composites. Additionally, the construction of magneto-oriented materials with two-dimensional particles is more conducive to the formation and stability of magneto-induced liquid crystals and photonic crystals. Previously, GO@Fe_3_O_4_ composites had several investigations in drug carriers [26], extraction [27,28], biosensors [29], photocatalysis [30], catalysts [31,32] and other applications, but research on the optical properties of GO decorated with uniform Fe_3_O_4_ has rarely been proposed. The loading state of Fe_3_O_4_ on the GO surface and the formation of Fe_3_O_4_ are the keys to affecting the properties of composites. To obtain composites with uniform loading of Fe_3_O_4_ on the GO surface, it was necessary to explore the effects of preparation conditions on the coating of Fe_3_O_4_ on the GO surface and the formation of Fe_3_O_4_.

In this paper, we took graphene oxide as the carrier and employed FeCl_3_·6H_2_O and FeSO_4_·7H_2_O as the iron source to prepare GO@Fe_3_O_4_ two-dimensional magnetically oriented nanocomposites using the co-precipitation method. The effects of process conditions on the effects of Fe_3_O_4_ coating on GO were explored, and two-dimensional nanocomposites with uniformly coated Fe_3_O_4_ on the GO surface were successfully obtained. A GO@Fe_3_O_4_ sol system was designed and constructed. The effects of magnetic field direction, magnetic field strength, and the solid content of GO@Fe_3_O_4_ sol on the optical properties of the GO@Fe_3_O_4_ sol such as liquid crystalline properties and photonic crystal properties were demonstrated and studied. The work provides a new research idea for obtaining magneto-optical nanomaterials with stable properties and high-dimensional structural units, which is a further step toward realizing its practical application.

## 2. Materials and Methods

### 2.1. Materials

FeCl_3_·6H_2_O (≥99.0% purity), FeSO_4_·7H_2_O (≥99.0% purity), hydrochloric acid (HCl, AR), and NH_3_·H_2_O (25–28% by wt) were all supplied by Sinopharm Chemical Reagent Beijing Co., Ltd. (Beijing, China). Graphene oxide (GO) was bought from XFNANO Tech. Co., Ltd. (Nanjing, China), whose diameter was 0.5–5 μm and thickness was 0.8–1.2 nm. The XRD and TEM images of GO are shown in Figure 1.

### 2.2. Preparation of GO@Fe_3_O_4_ Nanocomposites

The GO@Fe_3_O_4_ nanocomposites were prepared using the co-precipitation method. The preparation route of GO@Fe_3_O_4_ nanocomposites is shown in Figure 2. In a typical sample, firstly, graphene oxide (0.1 mg/mL, 20 mL) in water was sonicated for 30 min to form a stable dispersion and then 0.3 mmol FeCl_3_·6H_2_O was added to the graphene oxide (GO) dispersion and heated to 60 °C until the temperature was steady. Afterward, the mixture was stirred for 30 min and 0.15 mmol FeSO_4_·7H_2_O was mixed. In this instance, the Fe^3+^ and Fe^2+^ molar ratio should be 2:1 to obtain pure Fe_3_O_4_ [33] based on the reaction equation “Fe^2+^ + 2Fe^3+^ + 8OH^−^→Fe_3_O_4_ + 4H_2_O”. After vigorous stirring for 30 min, 0.5 mL NH_3_·H_2_O was injected slowly into the above hot mixture for reacting 30 min under stirring. The preparation of GO@Fe_3_O_4_ was carried out under a gentle nitrogen flow at 60 °C. After cooling to room temperature, a black material was precipitated and separated via a NdBFe magnet. The precipitate was washed with distilled water several times. The detailed formulas of these experiments with various experimental parameters are shown in Table 1. For comparison, Fe_3_O_4_ was also obtained without GO added to the dispersion.

### 2.3. Construction of GO@Fe_3_O_4_ Sol System

The GO@Fe_3_O_4_ nanocomposites were dispersed in aqueous solutions with pH of 0.8, 0.9, 1.0, 1.3, 1.5, and 1.6, respectively. The solid content in the aqueous solution was approximately 0.286 mg/mL (For GO). The obtained samples were stirred constantly and ultrasonically dispersed for 30 min. For comparison, the GO@Fe_3_O_4_ nanocomposites and GO were dispersed in distilled water (pH = 7).

### 2.4. Characterization

The XRD patterns of the samples were obtained using a D8 Advance X-ray powder diffractometer (Bruker, Germany) with filtered Cu Kα radiation operated at 40 kV and 40 mA. The diffraction data were collected from 3° to 70° and the resolution step size was 0.02°. Transmission electron microscope (TEM) images were taken on a Hitachi instrument (H-8100, Japan). For the TEM sample, a droplet of the solution was deposited onto a carbon-coated copper grid. FT-IR spectra of samples were obtained on a Spectrum 100 FT-IR spectrometer (Perkin-Elmer, American Fork, UT, USA), which recorded the wavenumber ranging from 400 to 4000 cm^−1^ with a KBr background. The magnetic characterization of the GO@Fe_3_O_4_ nanocomposites was conducted via a vibrating-specimen magnetometer (SQUID-VSM, Quantum Design, San Diego, CA, USA) at room temperature. The zeta potential data of the samples were recorded via a zeta potential analyzer (Zetasizer Nano ZS90, Malvern, UK).

The sample was dropped on a slide and its dispersion state was observed using an optical microscope (OM, 59XC, Shanghai optical instrument factory, Shanghai, China). The sample was placed in a glass capillary, under the polarized optical microscope (POM, 59XC, Shanghai optical instrument factory, Shanghai, China) with a constant magnetic field intensity of about 80 mT, by rotating the sample stage to change the angle between the direction of the magnetic field and the polarizer, and the liquid crystalline phases of the sample were simply visualized. The transmission spectra of GO@Fe_3_O_4_ sol were measured by simulating the POM characterization using a UV-Vis spectrophotometer equipped with two mutually perpendicular polarizers (Lambda 750, PerkinElmer, Waltham, MA, USA). NdFeB magnets were placed near the sample to provide the external magnetic field.

## 3. Results and Discussion

### 3.1. Preparation and Characterization of GO@Fe_3_O_4_ Two-Dimensional Nanocomposites

#### 3.1.1. XRD

To study the structure and composition of the GO@Fe_3_O_4_ composite prepared under various conditions, the GO@Fe_3_O_4_ obtained under various conditions, GO, and Fe_3_O_4_ were characterized by XRD. The results are displayed in Figure 3.

In Figure 3A–D, the characteristic peak positions of the GO@Fe_3_O_4_ and the Fe_3_O_4_ at about 2θ = 18.3°, 30.1°, 35.5°, 43.1°, 53.6°, 57.0°, and 62.6° are consistent with the standard card of the Fe_3_O_4_ (JCPDS 75-1609), demonstrating the formation of the Fe_3_O_4_ crystal. In addition, these characteristic peaks are broad and not very sharp, showing the crystal form of Fe_3_O_4_ was not particularly good. In Figure 3A, the intensity of these characteristic peaks gradually became stronger with the increase in the ferric salt dosages. S1 with a low ferric salt dosage showed the characteristic peak of the by-product FeOOH (JCPDS 76-0123) at 2θ = 40.0°. In Figure 3B, under various temperatures, these characteristic peak strengths of the samples were comparable, but S5 at a high temperature displayed the characteristic peak of FeOOH. In Figure 3C, S2 showed the formation of Fe_3_O_4_ without FeOOH, but S6 and S7 showed characteristic peaks of FeOOH. In Figure 3D, after various reaction times, the characteristic peak strengths of Fe_3_O_4_ in the samples were essentially the same, demonstrating that the reaction was sufficient at 15 min. As a result, the reaction conditions of 0.3–0.4 mmol Fe^3+^, 50–60 °C, and 0.5 mL ammonia water were conducive to the formation of Fe_3_O_4_ for this strategy. Additionally, there was a strong peak at 2θ = 11.18 ° in GO, which was attributed to the GO diffraction peak on the d001 crystal plane [34]. The characteristic peaks of the d001 crystal plane of the GO in all samples essentially vanished, indicating that GO was basically exfoliated into nanoflakes during the ultrasonic treatment. Therefore, it was concluded that the Fe_3_O_4_ and GO@Fe_3_O_4_ nanocomposites were successfully prepared.

#### 3.1.2. TEM

To investigate the loading state of Fe_3_O_4_ on the GO surface and the effects of ferric salt dosage, ammonia water dosage, temperature, and reaction time on the loading state of Fe_3_O_4_ on the GO surface, the samples prepared under various conditions were characterized by TEM. The results are displayed in Figure 4. Figure 4(A-1,A-2) display the TEM images of GO and Fe_3_O_4_ nanoparticles prepared without GO, respectively.

Figure 4(B-1–B-3) show the TEM results of the GO@Fe_3_O_4_ nanocomposites prepared with various ferric salt dosages. Figure 4B-1 shows a few small-scale GO nanoflakes without Fe_3_O_4_ coating and a few GO nanoflakes with Fe_3_O_4_ coating, demonstrating the coating of Fe_3_O_4_ was incomplete and the ferric salt dosage was insufficient. In Figure 4B-2, the Fe_3_O_4_ is uniformly coated on the GO surface without obvious Fe_3_O_4_ aggregations, showing that the ferric salt dosage was appropriate. S3 with a high ferric salt dosage in Figure 4B-3 displayed the aggregation of Fe_3_O_4_ on the GO surface, demonstrating the ferric salt dosage was excess. The TEM images of products obtained at various temperatures and with various ammonia water dosages are shown in Figure 4(C-1–C-3,D-1–D-3), respectively. In Figure 4(C-2,D-2), the Fe_3_O_4_ is uniformly coated on the GO surface, without apparent Fe_3_O_4_ aggregations. S4, S5, S6, and S7 suggested uneven loading of Fe_3_O_4_ nanoparticles on the GO surface. The TEM images of samples obtained after various reaction times are shown in Figure 4(E-1–E-3). The size and size distribution of Fe_3_O_4_ nanoparticles, in Figure 4E-2, were comparatively uniform and the size of Fe_3_O_4_ nanoparticles was about 10–25 nm, indicating that the nucleation time of Fe_3_O_4_ was appropriate. In Figure 4(E-1,E-3), the size of Fe_3_O_4_ nanoparticles in S8 and S9 was about 5–25 nm and 5–30 nm, respectively. S8 and S9 suggested the sizes of Fe_3_O_4_ nanoparticles and the loading state of Fe_3_O_4_ nanoparticles on the GO surface were uneven.

Therefore, under the optimized condition of 0.3 mmol Fe^3+^, 0.15 mmol Fe^2+^, and 0.5 mL ammonia water at 60 °C after 30 min reaction, the Fe_3_O_4_ nanoparticles were evenly coated on the GO surface and the size of Fe_3_O_4_ nanoparticles was uniform. The process conditions of S2 were used for subsequent discussions.

#### 3.1.3. FT-IR

To investigate the interaction mode between the Fe_3_O_4_ and GO in GO@Fe_3_O_4_, the GO, Fe_3_O_4_, and GO@Fe_3_O_4_ nanocomposites prepared under optimized conditions of S2 were characterized by FT-IR and the result is shown in Figure 5.

For GO@Fe_3_O_4_ and Fe_3_O_4_, the bands at 585 cm^−1^ and 586 cm^−1^ belonged to the stretching vibration of Fe-O, demonstrating the formation of Fe_3_O_4_ [35]. For GO, the bands at 1728 cm^−1^,1623 cm^−1^, and 1054 cm^−1^ were attributed to the stretching vibration of the C=O bond of carboxyl groups, the aromatic skeleton C=C stretching vibration of the unoxidized graphitic domains, and the vibrations of alkoxy C–O, respectively [36]. For GO@Fe_3_O_4_, the bands at 1728 cm^−1^ and 1054 cm^−1^ basically disappeared, due to the strong coordination effect between the Fe^3+^ and Fe^2+^ on the surface of Fe_3_O_4_ and the carboxylic groups in GO [35,37]. Additionally, the band at 1623 cm^−1^ shifted to 1592 cm^−1^. The strong coordination effect would decrease the vibrational frequency of the C=C band. These results of FT-IR confirmed the view that the Fe_3_O_4_ were adsorbed to the carboxylic groups on GO and Fe_3_O_4_ was successfully coated on the GO surface.

#### 3.1.4. VSM

The magnetic properties of GO@Fe_3_O_4_ obtained under optimized conditions of S2 and Fe_3_O_4_ were characterized by the vibrating-specimen magnetometer (VSM) at 300 K. As shown in Figure 6A, the saturation magnetization of the GO, GO@Fe_3_O_4_, and Fe_3_O_4_ was 0, 29.09 emu/g, and 37.32 emu/g, respectively. The coercive force of GO@Fe_3_O_4_ and Fe_3_O_4_ was 0 and they displayed superparamagnetism. The saturation magnetization of the GO@Fe_3_O_4_ nanocomposites was comparable to that of other reported magneto-oriented composites [38,39,40], although it was lower than that of Fe_3_O_4_. In Figure 6B, the GO@Fe_3_O_4_ nanocomposites could be easily and quickly attracted to the side of the container via magnets, illustrating the GO@Fe_3_O_4_ nanocomposites could respond quickly to the external magnetic field and the GO@Fe_3_O_4_ nanocomposites were adequate to meet the requirements of magneto-oriented materials.

### 3.2. Construction of the GO@Fe_3_O_4_ Sol System

The GO@Fe_3_O_4_ sol system was constructed under various pH conditions. The stability of each sample was characterized by an optical microscope to study the effect of pH on the stability of the samples. The results are shown in Figure 7.

In Figure 7A, the GO@Fe_3_O_4_ nanocomposites were dispersed in distilled water (pH = 7), and numerous black agglomerated particles were observed, which was detrimental to the optical properties of materials [24,41]. Here, we used the electrostatic stabilization mechanism to add some hydrochloric acid to enhance the dispersion of the GO@Fe_3_O_4_ nanocomposites. In Figure 7B–G, as the pH decreased, the number of black agglomerated particles in the samples first decreased and then increased. The GO@Fe_3_O_4_ nanocomposites displayed uniform dispersion in Figure 7C–E. Additionally, the zeta potentials of these samples were higher than that of other pH conditions as shown in Table 2, indicating that the GO@Fe_3_O_4_ nanocomposites were more evenly dispersed. The stable GO@Fe_3_O_4_ sol system could be obtained via keeping pH = 1.0–1.5.

### 3.3. The Properties of the GO@Fe_3_O_4_ Two-Dimensional Nanocomposites

#### 3.3.1. Magnetic Orientation Properties

The magnetic orientation properties of GO@Fe_3_O_4_ sol were evaluated using an optical microscope and transmission electron microscope under a magnetic field. The results are displayed in Figure 8 and Figure 9. The strength of the magnetic fields used in this work was fixed.

In Figure 8A-2, after the magnetic field was applied, the black agglomerates in the sample were attracted aside by the magnetic field force, indicating that the force between the GO@Fe_3_O_4_ nanoflakes was insufficient to balance the magnetic field force. For the stable GO@Fe_3_O_4_ sol, after the magnetic field was applied in Figure 8(B-2–D-2), the GO@Fe_3_O_4_ clearly was oriented into an ordered structure and arranged along the direction of the magnetic field, demonstrating the force between GO@Fe_3_O_4_ nanoflakes could balance the magnetic field force.

The GO@Fe_3_O_4_ sol (pH = 1.3) was also characterized by TEM to study the microscopic magnetic orientation properties. In Figure 9A, without the magnetic field, the GO@Fe_3_O_4_ nanocomposites were chaotic and disordered. In Figure 9B, the GO@Fe_3_O_4_ nanoflakes could be arranged into an ordered structure along the direction of the magnetic field under the influence of the external magnetic field.

#### 3.3.2. Liquid Crystalline Properties

The liquid crystalline properties of GO@Fe_3_O_4_ sols were evaluated by observing the birefringence of the GO@Fe_3_O_4_ sol sealed in a capillary using the polarized optical microscope (POM). The solid content of GO@Fe_3_O_4_ sol was 0.286 mg/mL. Figure 10 displays the schematic diagram for the POM characterization. The results are shown in Figure 11 and Figure 12.

Since the orientation direction of GO@Fe_3_O_4_ nanoflakes was directly influenced by the magnetic field direction, altering the magnetic field direction was equivalent to altering the light direction. For the GO@Fe_3_O_4_ sol (pH = 1.3) in Figure 11, when the angle between the polarizer and the magnetic field direction was 0° or 90°, in Figure 11A-1 or Figure 11A-3, it became completely dark. When the sample was rotated to 45° or 135° in Figure 11A-2 or Figure 11A-4, a single birefringent phase could be observed and the strongest transmitted light was visible. In the hydrochloric acid system, H+ would be adsorbed to the surface of the GO@Fe3O4 nanoflakes to form the electric double layer, which would cause strong repulsion between the nanoflakes. In the stable GO@Fe_3_O_4_ sol system, the electric double-layer force, gravity, and van der Waals forces were in balance so that the GO@Fe_3_O_4_ nanoflakes were uniformly dispersed. After the magnetic field was applied, the GO@Fe_3_O_4_ nanoflakes formed an ordered arrangement with the head and tail connected along the magnetic field direction [42]. Interestingly, the reversible magnetic response of the GO@Fe3O4 sol was observed in Figure 11(B-1–B-4). In Figure 11B-1, without the magnetic field, the sample was dark and the system was isotropic. After the magnetic field was applied for about 0.5 s, Figure 11B-2 showed a bright route. The route could be repeatedly switched between light and dark by repeatedly adding and removing the magnet, and the switching time was within 0.5 s in Figure 11(B-3,B-4). The adjustment was transient and reversible and could be reusable many times repeatedly (see Appendix A for details). The magnetic field was crucial in the formation of ordered structures and nematic phases in the GO@Fe_3_O_4_ sol. Additionally, the liquid crystalline phases were also seen in the GO@Fe_3_O_4_ sol systems of pH = 1.0 and pH = 1.5 in Figure 12(A-1,A-2). As a result, the stable GO@Fe_3_O_4_ sol systems could exhibit liquid crystalline properties under the effect of the magnet field. The POM images of GO@Fe_3_O_4_ sol and GO under the pH = 7.0 condition are shown in Figure 12(A-3,A-4), respectively. By turning the magnetic field direction, the birefringence of both samples was not seen, demonstrating that neither system was capable of aligning with the external field to form the liquid crystalline phases.

Additionally, using the UV-Vis spectrophotometer to characterize the light transmittance of GO@Fe_3_O_4_ sol at an angle of 45° between the polarizer and the magnetic field direction, the relationship between the magnetic orientation properties of the GO@Fe_3_O_4_ sol and the magnetic field strength was revealed. The solid content of GO@Fe_3_O_4_ sol was 0.286 mg/mL. The magnetic field strength was adjusted by changing the distance between the sample and the magnet. Figure 13A,B depict the experimental setup and results, respectively. In Figure 13B, without the magnetic field, the GO@Fe_3_O_4_ sol was isotropic and showed a light transmittance of 0. When the magnetic field was applied, with the magnet approaching the sample from 3 cm to 2 cm to 1 cm, the maximum light transmittance of the sample was 0.83%, 0.91%, and 1.15%, respectively. As the magnetic field strength increased, the orientation properties of the sample would be improved, and the structure’s order also increased. This would permit more light to pass through and result in higher light transmittance.

The aforementioned phenomenon suggested that after the Fe_3_O_4_ was coated on the GO surface, the GO@Fe_3_O_4_ nanoflakes could be aligned with the external field to form the liquid crystalline phases, which could be regulated by the magnetic field. The regulation was instant, reversible, and repeatable, which overcame the problems of the long corresponding period and the difficulty of regulating for general inorganic minerals.

#### 3.3.3. Photonic Crystal Properties

The reflection color and intensity of GO@Fe_3_O_4_ sol for visible light were observed to investigate the photonic crystal properties of GO@Fe_3_O_4_ sol. The impacts of the magnetic field intensity, direction, and the solid content of GO@Fe_3_O_4_ sol on photonic crystal properties were studied. The results are shown in Figure 14 and Figure 15.

The Bragg reflections of the sample illuminated by a strong natural light were seen under various magnetic field strengths. The magnetic field strength was changed by varying the distance between the magnet and the sample in Figure 14A-1–A-3. In Figure 14A-1, without the magnetic field, the GO@Fe_3_O_4_ nanoflakes in the sample were disordered and the colored reflection could not be seen. After the magnetic field was applied in Figure 14A-2,A-3, the obvious yellow reflection could be discerned, and it became brighter as the magnetic field strength increased. As Lagerwal [43] discussed, the periodicity of the photonic crystal created the photonic bandgap resulting in a strongly colored reflection, which required optical path difference (=d·sinθ) on the order of λ (wavelength) in Figure 14B. Therefore, under the influence of the magnetic field, the GO@Fe_3_O_4_ nanoflakes were aligned with the direction of the magnetic field, providing a long-range ordered arrangement for the structure. The period (d) of the GO@Fe_3_O_4_ nanoflakes was comparable to the wavelength of yellow light. The greater the magnetic field strength, the better the ordering degree of GO@Fe_3_O_4_ nanoflakes, and the stronger the Bragg reflections.

The influence of magnetic field direction on Bragg reflection was investigated and is shown in Figure 15A-1–A-3. The magnetic field direction was regulated by changing the placement of the magnet. In Figure 15A-1, when the magnetic field direction was parallel to the desktop, a bright-yellow line could be seen in the middle of the sample. In Figure 15A-2, the magnetic field direction was perpendicular to the desktop, and there were no color reflections in the entire sample. While the magnet was on the side of the sample in Figure 15A-3, the bright-yellow reflection light was also observed. It demonstrated the magnetic field direction could alter the orientation of the GO@Fe_3_O_4_ nanoflakes and determine whether the Bragg reflection could be formed. Additionally, Figure 15B-1–B-4 depict the Bragg reflection of the GO@Fe_3_O_4_ sol with various solid contents. As the solid content decreased, while keeping the other parameters constant, the bright-yellow reflection line in the middle of the samples became shallower and shallower until it was difficult to see. The decreasing of the solid content increased the interlayer spacing between the oriented GO@Fe_3_O_4_ nanoflakes, increased light penetration, and resulted in weaker reflection. As a result, a higher solid content of GO@Fe_3_O_4_ sol was crucial for the Bragg reflection.

The solid content with GO@Fe_3_O_4_ nanoparticles as building blocks displayed color reflection at 0.00572 mg/mL in Figure 16, and a uniform blue region could be seen clearly. The orientation of the sample was consistent with the magnetic field direction, indicating that the color reflection was caused by the ordered structure. This phenomenon showed that the GO@Fe_3_O_4_ was oriented to form an ordered structure under the influence of the magnetic field and that the spacing of the ordered structure was in the visible-wavelength range, leading to reflection in the visible region.

In the GO@Fe_3_O_4_ sol system, the reflection of GO@Fe_3_O_4_ sol was limited and influenced by the magnetic field strength, magnetic field direction, and solid content of the GO@Fe_3_O_4_ sol. The stronger the strength of the magnetic field, the better the orientation of the GO@Fe_3_O_4_ nanoflakes, and the stronger the reflective intensity. When the magnetic field direction was changed, different reflection regions could be seen. Additionally, the reflection intensity decreased with decreasing solid content. In the GO@Fe_3_O_4_ sol system, the magnetic field force was determined by the magnetic field strength and direction, and the interparticle force was influenced by the solid content. The two forces together determined the spacing of the orientation structure [44], which impacted the reflection effect of the structure on visible light.

## 4. Conclusions

In summary, GO@Fe_3_O_4_ two-dimensional magnetically oriented nanocomposites were prepared successfully via the co-precipitation method, and Fe_3_O_4_ nanoparticles with uniform size were coated evenly on the GO surface under the appropriate experimental conditions. The GO@Fe_3_O_4_ nanocomposites were superparamagnetic and had a saturation magnetization of 29.09 emu/g, whose coercive force was 0. The stable GO@Fe_3_O_4_ sol system was constructed by dispersing GO@Fe_3_O_4_ nanocomposites in diluted hydrochloric acid. The GO@Fe_3_O_4_ nanoflakes demonstrated magnetic-field-assisted self-assembly into liquid crystalline phases and photonic crystals. The optical properties of GO@Fe_3_O_4_ sol could be controlled by the magnetic field direction, the magnetic field strength, and the solid content of the GO@Fe_3_O_4_ sol, and it was reversible and transient. The GO@Fe_3_O_4_ two-dimensional magnetically oriented nanocomposites with stable properties and high-dimensional structural units obtained in this work provide a fresh idea for photonic switch materials, anti-counterfeiting materials, and gas barrier materials.

## Figures and Tables

**Figure 1 materials-16-00476-f001:**
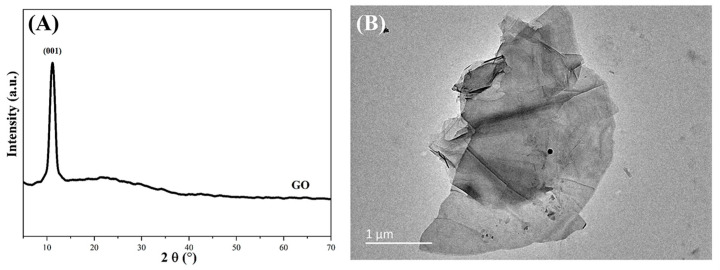
(**A**) XRD and (**B**) TEM images of GO.

**Figure 2 materials-16-00476-f002:**
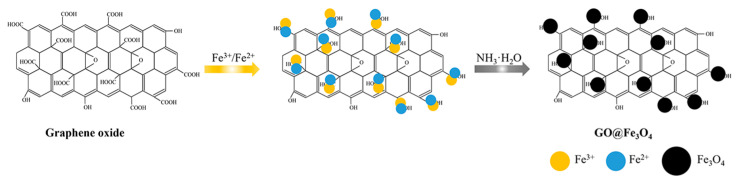
The preparation route of GO@Fe_3_O_4_ nanocomposites.

**Figure 3 materials-16-00476-f003:**
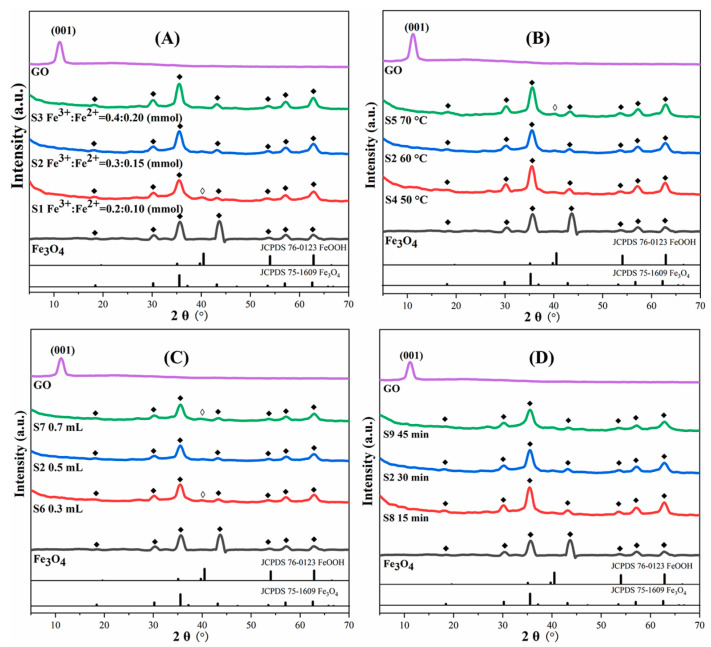
XRD patterns of GO and GO@Fe_3_O_4_ prepared under various conditions (◆: Fe_3_O_4_ ◇: FeOOH). (**A**) various ferric salt dosages; (**B**) various temperatures; (**C**) various ammonia water dosages; (**D**) various reaction times.

**Figure 4 materials-16-00476-f004:**
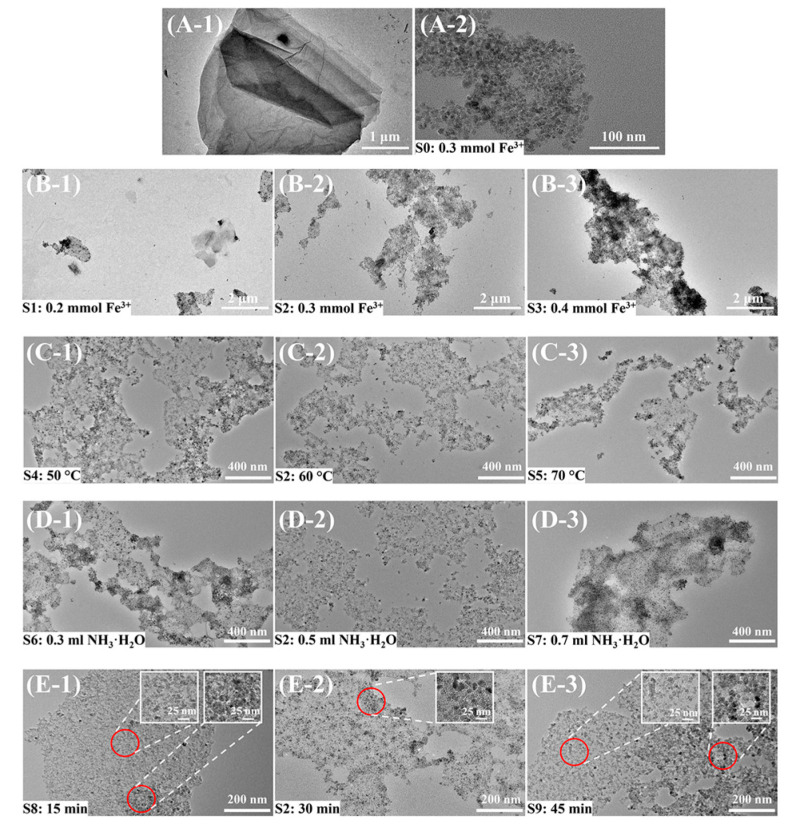
TEM images of GO, Fe_3_O_4_, and GO@Fe_3_O_4_ prepared under various conditions. (**A-1**) GO, (**A-2**) Fe_3_O_4_; (**B-1**–**B-3**) various ferric salt dosages; (**C-1**–**C-3**) various temperatures; (**D-1**–**D-3**) various ammonia water dosages; (**E-1**–**E-3**) various reaction times.

**Figure 5 materials-16-00476-f005:**
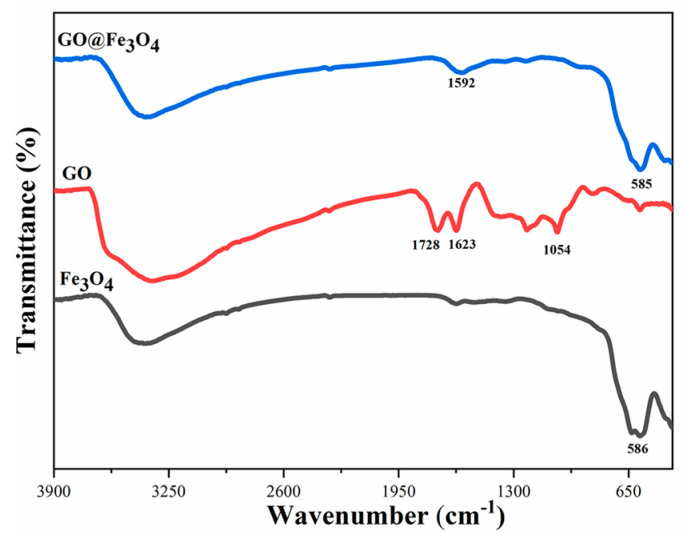
FT-IR patterns of Fe_3_O_4_, GO and GO@Fe_3_O_4_.

**Figure 6 materials-16-00476-f006:**
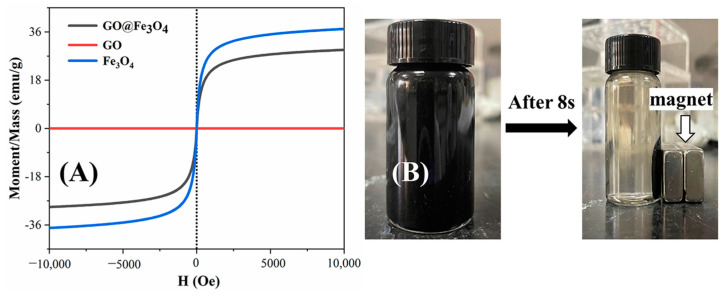
(**A**) Magnetization curves of GO, Fe_3_O_4_, and GO@Fe_3_O_4_ at 300 K. (**B**) Real image of magnet separation of the GO@Fe_3_O_4_ nanocomposites.

**Figure 7 materials-16-00476-f007:**
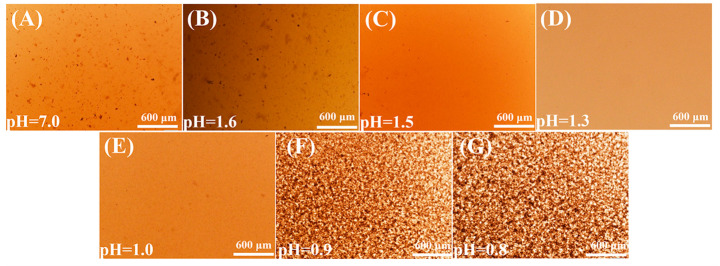
Optical microscope images of GO@Fe_3_O_4_ sol under various pH conditions (**A**) pH = 7.0; (**B**) pH = 1.6; (**C**) pH = 1.5; (**D**) pH = 1.3; (**E**) pH = 1.0; (**F**) pH = 0.9; (**G**) pH = 0.8.

**Figure 8 materials-16-00476-f008:**
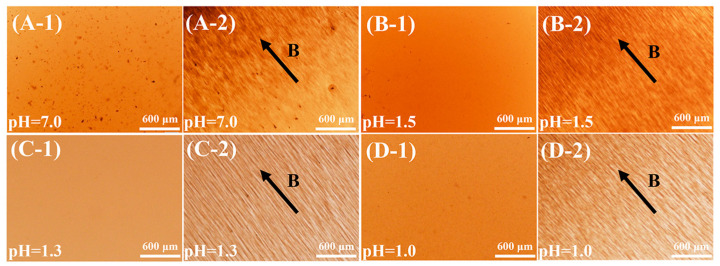
Optical microscope images of the samples (**A-1**–**D-1**) without the magnetic field and (**A-2**–**D-2**) with a magnetic field. (The B shows the direction of the magnetic field).

**Figure 9 materials-16-00476-f009:**
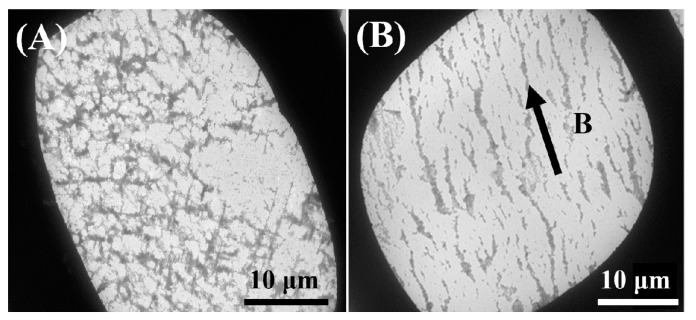
TEM images of the sample (**A**) without the magnetic field and (**B**) with the magnetic field. (The B shows the direction of the magnetic field).

**Figure 10 materials-16-00476-f010:**
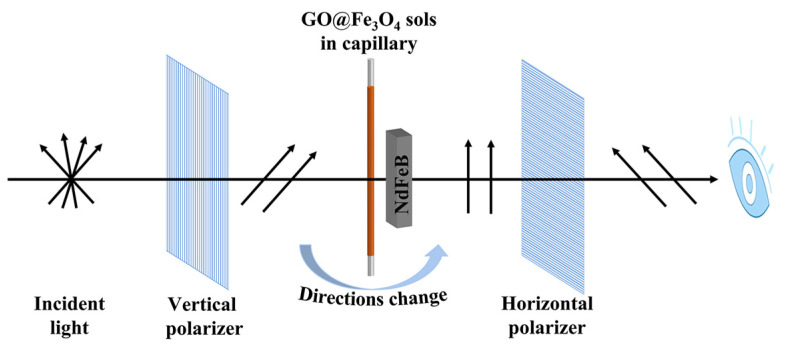
Schematic diagram for the POM characterization.

**Figure 11 materials-16-00476-f011:**
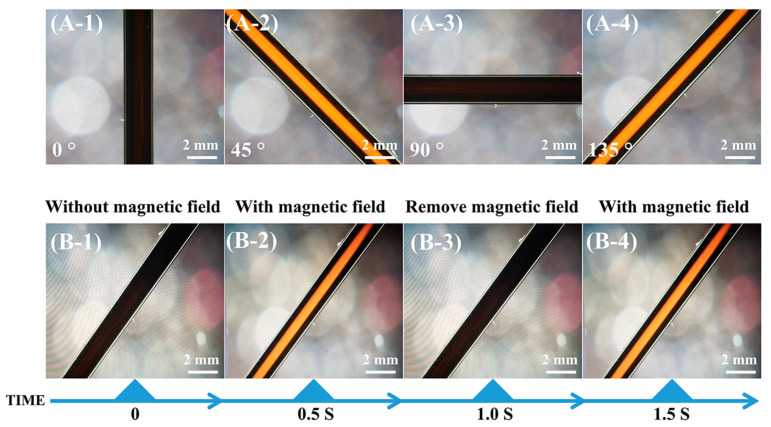
(**A**) Magnetic-field-direction-dependent polarized optical microscope (POM) of the aligned system showing uniform extinction and passage of light; (**A-1**) the angle between the polarizer and the magnetic field direction was 0°; (**A-2**) 45°; (**A-3**) 90°; (**A-4**) 135°. (**B-1**–**B-4**) POM images of samples: (**B-1**) without a magnetic field, (**B-2**) with a magnetic field, (**B-3**) after removal of the magnetic field, (**B-4**) with a magnetic field again.

**Figure 12 materials-16-00476-f012:**
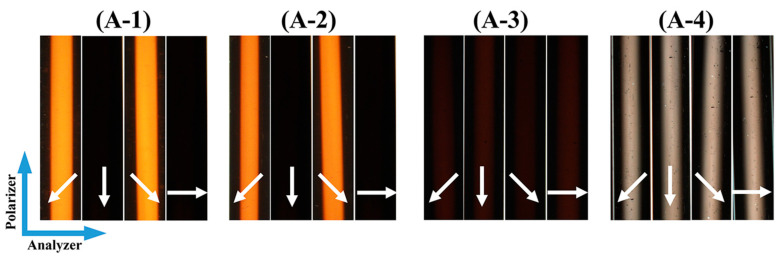
POM images of the GO@Fe_3_O_4_ sol in various pH conditions and GO aqueous solution under the magnetic field in various directions (the white arrows indicate the magnetic field direction). (**A-1**) GO@Fe_3_O_4_-pH = 1.0, (**A-2**) GO@Fe_3_O_4_-pH = 1.5, (**A-3**) GO@Fe_3_O_4_-pH = 7.0, (**A-4**) GO-pH = 7.0.

**Figure 13 materials-16-00476-f013:**
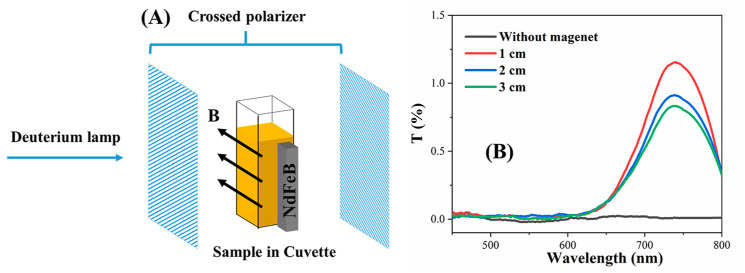
(**A**) The experimental setup for measuring light transmittance using the spectrophotometer; (**B**) Transmittance of light through two orthogonal polarizers and GO@Fe_3_O_4_ sol at various magnetic field intensities. (The B shows the direction of the magnetic field).

**Figure 14 materials-16-00476-f014:**
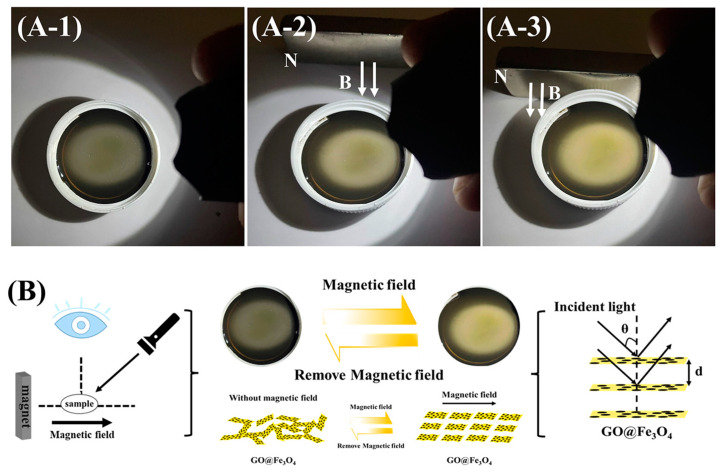
(**A-1**–**A-3**) Real images of the Bragg reflections of GO@Fe_3_O_4_ sol under various magnetic field strengths and (**B**) Schematic diagram of the orientation of GO@Fe_3_O_4_ nanoflakes and the formation of Bragg reflections. (**A-1**) without magnet; (**A-2**) the distance between the sample and magnet was 2 cm; (**A-3**) the distance between the sample and magnet was 0.2 cm. (The B shows the direction of the magnetic field).

**Figure 15 materials-16-00476-f015:**
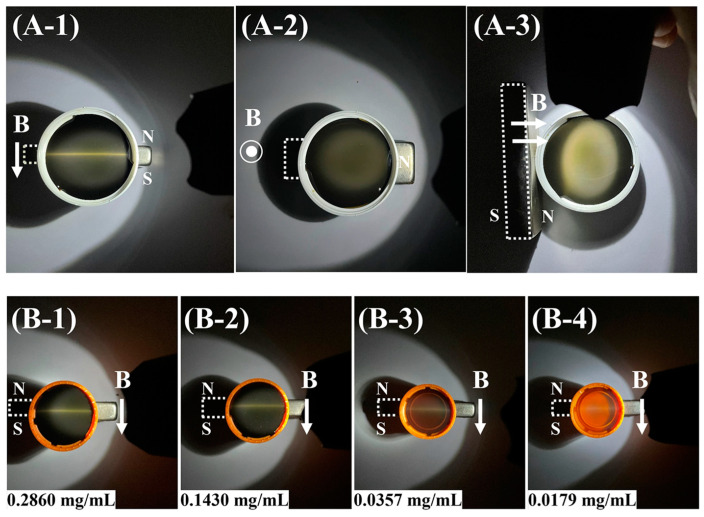
Real images of the GO@Fe_3_O_4_ sol Bragg reflections (**A-1**–**A-3**) under various magnetic field directions and (**B-1**–**B-4**) with various solid content of GO@Fe_3_O_4_ sol in the magnetic field. (The B shows the direction of the magnetic field).

**Figure 16 materials-16-00476-f016:**
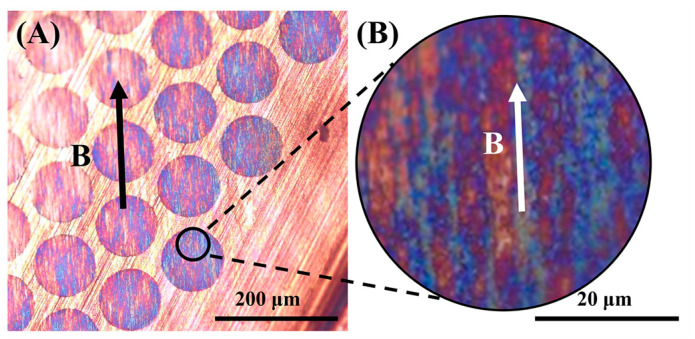
(**A**) Optical microscope images of aligned GO@Fe_3_O_4_ nanoparticles on the substrate; (**B**) the enlarged view in the black circle of (**A**). (The B shows the direction of the magnetic field).

**Table 1 materials-16-00476-t001:** Experimental parameters for the preparation of all the samples.

Sample	GO (mg)	FeCl_3_·6H_2_O (mmol)	FeSO_4_·7H_2_O (mmol)	NH_3_·H_2_O (mL)	Temperature (°C)	Reaction Time (min)
S0	0	0.3	0.15	0.5	60	30
S1	2	0.2	0.10	0.5	60	30
S2	2	0.3	0.15	0.5	60	30
S3	2	0.4	0.20	0.5	60	30
S4	2	0.3	0.15	0.5	50	30
S5	2	0.3	0.15	0.5	70	30
S6	2	0.3	0.15	0.3	60	30
S7	2	0.3	0.15	0.7	60	30
S8	2	0.3	0.15	0.5	60	15
S9	2	0.3	0.15	0.5	60	45

**Table 2 materials-16-00476-t002:** Zeta potential of the GO@Fe_3_O_4_ sol under various pH conditions.

pH	7.0	1.6	1.5	1.3	1.0	0.9	0.8
Zeta potential (mV)	18.10	38.37	39.90	39.47	39.03	38.50	38.03

## Data Availability

Data are contained within the article.

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
