# Peer review of "Study on the Preparation and Optical Properties of Graphene Oxide@Fe3O4 Two-Dimensional Magnetically Oriented Nanocomposites"

_materials, 2023, doi:10.3390/ma16020476_

Round 1

Reviewer 1 Report

The authors have synthesized the of GO@Fe3O4 composites using co-precipitation approach and investigate the phase formation, morphology, physical properties of these composite samples. As a whole, the topic is interesting and is suitable for publication in Materials. Here there are some issues to be done in the revision of the manuscript before publishing:

1.     Some references about the previous topics with various application may be read during the revision and might be cited in the revision to extend the readership, e.g. DOI: 10.1016/j.ceramint.2022.03.090 ; 10.1016/S1872-5813(21)60063-4 ; 10.1016/j.ceramint.2021.08.083

2.     Author the carry out the synthesis with variations of ferric salt dosages, temperatures, ammonia water dosages and reaction times. What is the reason for the researcher to do this variation? This has not been seen in the introduction.

3.     The author performed TEM analysis to observe the distribution of Fe3O4 in the GO layer, why the author does not discuss the nanoparticle size of Fe3O4 from this analysis (showing the zoomed TEM image for the particle). I think the topic of nanomaterials will give impact on paper quality.

4.     It is interesting that since the composite exhibits the possibility of separation from solution by using an external magnet. The authors may be able to show the remnant magnetization value of the enlarged curve on the low magnetic field portion of the inset figure.

5. I suggest the author to revise the figure of magnetization curve, using the magnetic field unit Oersted/Oe (commonly used international unit conversion).

6.     The author can add a discussion regarding the magnetic properties of this composite for reusable use in discussion.

7.     The author shows analysis data from several variations of the sample, but in the FTIR, VSM and so on, only 1 data is displayed, which is referred to as S2. Which part of the sample is S2 because in the XRD data all samples are abbreviated as S2? Please explain in the response.

Reviewer 2 Report

This paper has prepared graphene oxide@Fe3O4 nanocomposite using co-precipitation method and characterized using XRD, FTIR, TEM and microscopic techniques. The condition for stable and uniformly coated Fe3O4 on GO has been shown. Under magnetic field, the nanoflakes attain liquid crystalline phase that shows an interesting magnetic field dependent optical property. However, several discussions in this manuscript need to be clarified and necessary characterizations are required to support the conclusion.

1.      In figure 2 author has shown that the Fe 2+/Fe3+ bind to the Carbon center of GO. How does the bond formation happen? According to the FTIR the Fe2+/Fe3+ bonds with the carboxylic acid not with the C=C .

2.      In the XRD section author claims that the crystal form of the GO@Fe3O4 is “better”. Please explain how it is better. Also, from the XRD it looks like the peaks are broad and not very sharp. Explain this. Please add the XRD of just Fe3O4 that is prepared in this work and compare with GO@Fe3O4.

3.      Authors show at higher temperature FeOOH formation is higher? Any particular reason why higher temperature favors FeOOH?

4.      According to the authors, sample S2 shows no FeOOH formation where 0.5 ml NH3 is used, whereas sample S6 and S7 show FeOOH formation with 0.3 and 0.7 ml of NH3 respectively. Any reason why FeOOH formation occur with lower and higher NH3. What is so special with 0.5 ml NH3?

5.      From the XRD data, author claims the GO exfoliated into nanoflakes during the GO@Fe3O4 process. Did it actually form the nanoflakes during the Fe3O4 formation process or does it  actually  form during sonication of GO and FeCl3 addition. A XRD of the solution before addition of FeCl2 will be useful. This will show whether Fe3O4 forms separately and then form the complex or it actually forms insitu on the surface of the GO.

6.      TEM image figure 4(B-1) shows small scale GO without Fe3O4 coating. The XRD data with the same Ferric Chloride concentration show GO@Fe3O4 composite formation. Explain this anomality?

7.      Sample S3 with higher Ferric chloride leads to aggregation. Why this happens?

8.      Provide the FTIR of the GO+FeCl3 solution (before adding FeCl2). This will show how Fe3+ bind the GO surface.

9.      Author shows as the PH decreases the number of agglomerated particles first decreases and then increases. Why does it increase when PH decreases further.

10.   For the magnetic field dependent optical microscope experiment (POM), what kind of concentration has been used for the experiment. Also, what kind of concentration has been used for the magnetic field dependent light transmission experiments. For both the experiments why the nanocomposites did not separate like it happened in figure 6. Need further clarification.

1.   Author explained at lower pH, H+ would be adsorbed on the surface of the nanoflakes and will cause repulsion between nanoflakes. And the applied magnetic field will compensate this force to make the nanoflakes parallel to the magnetic field. This part is little confusing. The magnetic field will act on the nanoflakes as a whole and the van der waals force will be experienced between the nanoflakes. How can these forces balance each other as the direction is not opposite. This part needs some clarification.

Round 2

Reviewer 1 Report

I am happy that the authors have revised their manuscript in line with all reviewer comments and that it is now suitable for publication.